# Silver Catalyzed Site-Selective C(sp^3^)−H Bond Amination of Secondary over Primary C(sp^3^)−H Bonds

**DOI:** 10.3390/molecules27196174

**Published:** 2022-09-21

**Authors:** Luzhen Jiao, Dawei Teng, Zixuan Wang, Guorui Cao

**Affiliations:** State Key Laboratory Base of Eco-Chemical Engineering, College of Chemical Engineering, Qingdao University of Science and Technology, Qingdao 266042, China

**Keywords:** silver-catalyzed nitrene, intramolecular selective amination, cyclic sulfamate derivatives, primary C(sp^3^)−H bond

## Abstract

Sulfamates are widespread in numerous pharmacologically active molecules. In this paper, Silver/Bathophenanthroline catalyzed the intramolecular selective amination of primary C(sp^3^)−H bonds and secondary C(sp^3^)−H bonds of sulfamate esters, to produce cyclic sulfamates in good yields and with a high site-selectivity. DFT calculations revealed that the interaction between sulfamates and **L10** makes the molecule more firmly attached to the catalyst, benefiting the catalysis reaction. The in vitro anticancer activity of the final products was evaluated in MCF-7 breast cancer cells.

## 1. Introduction

Sulfamates not only form the core of natural products but are essential scaffolds for the development of medicinal chemistry [1,2,3,4,5]. In fact, the importance of sulfamate in the pharmacophores can be gleaned from its appearance in biologically and pharmacologically significant compounds, such as 2-Alkylpyrrole sulfamates isolated from the marine worm *Cirriformia tantalate* [6], Avasimibe as an inhibitor of acyl coenzyme A:cholesterol acyltransferase (ACAT) [7], and the new drug Topiramate with anticonvulsant effects [8], as well as many others [9]. More and more cyclic sulfamates have been synthesized, and they have exhibited promising bioactivities, such as Haplosamates with HIV inhibitory activity [10], GABA_A_ receptor inhibitor [11], and calcium-sensing receptor agonists (Figure 1) [12].

Consequently, there have been a variety of strategies developed for the synthesis of cyclic sulfamates, such as intramolecular aziridination reaction [13], hydrogenation of cyclic sulfamate imines [14], alkyne metathesis [15], cyclizations of amino alcohols [16], and most recently, metal-nitrenoid C–H insertions [17,18,19,20]. Among these methods, direct selective amination of inert aliphatic C(sp^3^)–H bonds, which exists widely in nature, not only meets the requirements of “atomic economy”, but is the most simple and efficient strategy. Inexpensive metal-catalyzed nitrene transfer reactions have become an effective C-N bond formation method, following the contributions of Schomaker [21,22,23,24,25,26,27,28], Liu [29,30,31], White [32,33,34], Zhang [35,36,37,38], Che [39,40,41], and others [42,43,44,45,46].

To date, the selective amination of aliphatic C(sp^3^)–H bonds has mainly been limited to the site-selectivity of unactivated substrates (e.g., tertiary alkyl C–H bonds and benzylic γ-C–H bonds) and activated substrates (e.g., allylic and benzylic C–H bonds)23, [25,47,48,49,50,51,52,53,54,55,56]. Due to the similar high bond-dissociation energy of aliphatic C-H bonds, only a handful of studies have been conducted to identify the selective amination of aliphatic C(sp^3^)–H bonds. Pioneering work by Schomaker et al. successfully controlled the selective amination of secondary C(sp^3^)−H bonds (activated substrates) and tertiary C(sp^3^)−H bonds (Figure 1a) [57]. In addition, the Du Bois group also studied a substrate containing multiple reaction sites (benzylic C–H bonds and secondary C–H bonds), to investigate the effect of substrate and catalyst structure on amination (Figure 1b) [58]. Recently, Liu developed an iron-catalyzed selective amination of unactivated substrates, but only demonstrated a 2.5:1 site-selectivity toward the amination (Figure 1c) [59]. Although those examples represent powerful methods, based on the challenge of inert C–H bond activation, more catalytic systems need to be developed to study the selective amination of unactivated substrates.

To address this challenge, and aiming to explore the selective amination of secondary C(sp^3^)−H bonds versus primary C(sp^3^)−H bonds, we report herein that a series of novel sulfamates containing quaternary carbon centers were catalyzed using silver-complex, to obtain cyclic sulfamates with a high yield and good site-selectivity (Figure 1d). Moreover, a computational study of ligand-substrate steric repulsions indicated the reason for selective amination. The resulting structures possessed cyclic sulfamate fragments and exhibit potential antitumor activities.

## 2. Results and Discussion

At the outset of our study, 2-methyl-2-phenylpropyl sulfamate ester **1** was selected as the representative substrate to screen the optimal Ag/ligand (Table 1). We found that unsubstituted bipyridine (**L1**) or bipyridine ligand bearing trifluoromethyl group (**L2**) are inferior to electron-donating bipyridine derived ligands (**L3**–**L4**), reflecting that the effect of the electron-donating ligands on the reaction outcome is beneficial (Entries 1–4). Ligand **L5** with a large steric hindrance (Entry 5) could also catalyze the reaction with moderate yields (47%) and poor site-selectivity (5.6:1). Given that the substituted bis(oxazoline) ligand (dmbox) are proven catalysts for the C–H amination of the γ-C–H bond^28^, we then turned our attention to the examination of the dmbox ligand (entry 6). Unfortunately, a trace product was obtained in the case of using the dmbox ligand. Screening of ligand effects (Entries 7−11) revealed that phenanthroline ligands had a better response to the intramolecular selective amination of sulfamate esters. Notably, **L10**-catalyzed C-H selective amination resulted in exclusive formation of cyclic sulfamate **1a** in a 53% yield and more than 15:1 site-selectivity, which indicates a remarkably high level of reactivity for aliphatic C(sp^3^)-H bond amination with this catalyst (Entry 10). Considering that the pyridine ligand may play a key role in this reaction, the Schiff base ligand and terpyridine ligands were examined. Screening of ligand effects (Entries 12−14) revealed that the Schiff base ligand **L12** and terpyridine ligands **L13–L14** had a poor response to the intramolecular selective amination of sulfamate esters. To summarize, the enhanced reactivity of the C-H amination reaction may be attributed to a difference of steric hindrance and electronic properties between these ligands, as evidence suggests that **L10** is a significantly better ligand and gave the best results, on the basis of the reaction yield and site-selectivity.

The temperature did not have a significant impact on the site-selectivity but impacted the yields (Table 2 Entries 1−7). The screening of several different temperatures revealed that 55 °C was more suitable for the reaction, with a 72% yield and more than 15:1 site-selectivity being achieved (Entry 3). Among the tested metal salts, AgBF_4_ and AgSbF_6_ had poor catalytic effect on selective amination (Entry 8 and 11). Moreover, AgClO_4_ gave the best results, in terms of the reaction yield and site-selectivity.

After determining the optimal catalytic conditions, we compared this catalytic system with other catalytic systems reported in the literature for selective amination (Table 3). The catalyst of Fe(OTf)_2_ and [Rh(OAc)_2_]_2_ gave **1a** with high yield but a markedly reduced site-selectivity (Entry 2 and 4). However, the intramolecular selective amination could not be catalyzed by [Fe^III^(Pc)]SbF_6_ and Cu(OTf)_2_ (Entry 3 and 5).

Under the optimized conditions, we explored the scope of this silver catalysis, employing a broad variety of unactivated substrates in intramolecular selective amination (Figure 2). To our delight, sulfamates with electron-donating and electron-withdrawing groups on the phenyl ring all served as excellent substrates for the selective amination, in up to 80% yield and >15:1 site-selectivity (**1–5**). Unfortunately, the diastereoselectivity of sulfamates bearing meta- and para-substituents on phenyl groups was poor, affording the cyclic sulfamates 1.8:1 to 3.8:1 dr (**3–5**). The 2-methyl-2-phenylpentyl sulfamate (**6**) exhibited good reactivity, forming the corresponding cyclic sulfamate in 70% yield and >15:1 site-selectivity. Surprisingly, sulfamates containing a large steric hindrance (**7** and **8**) could also obtain cyclic sulfamates, in up to 75% yield and >15:1 site-selectivity.

Given the high reactivity of the 2-methyl-2-phenylbutyl sulfamate **1**, we next examined the scope of this selective amination with regard to the substitution at the secondary C(sp^3^)-H bond (Figure 3). The effect of the electron-donating group on the secondary C(sp^3^)-H bond was explored with **9**–**10**. Generally, good to excellent yields and site-selectivities were obtained in the presence of electron-donating substituents on secondary C(sp^3^)−H bonds (**9–10**). Next, differences in the preference for amination of propargylic and allylic C-H bonds over primary C(sp^3^)-H bonds using **L10**/AgClO_4_ were briefly explored (**11–12**). As expected, substrates containing a propargylic substituent showed an improved preference for insertion at the allylic C-H bond, activated by a neighboring π-system. Regrettably, the expected results were not obtained in **13**.

A possible reaction pathway was proposed (Figure 4). Treatment of substrate **1** with PhIO generates iminoiodinane, which reacts with the silver catalyst, leading to the formation of an metallonitrene species together with iodobenzene. Then, direct C-H insertion or H-atom abstraction/radical recombination of oxathiazinanes yielded the desired product and regenerated the catalyst [23]. Importantly, the C-H bond cleavage was calculated as the HAT step, with TS-methylene having a 3.06 kcal/mol lower free energy than TS-methyl.

We constructed a computational model to better understand the catalytic system (Figure 5). The sulfamate was close to the ligand scaffold, with a nitrogen atomethylene distance of 4.14 Å. Conversely, the distance from the nitrogen atom to the terminal methyl was 5.21 Å. Therefore, amination of methylene is likely to occur more easily than amination of the methyl group. Furthermore, the interaction between sulfamates and **L10** makes the molecule more firmly attached to the catalyst, benefiting the catalysis reaction. It is thought that the excellent site selectivity is a result of the methylene group being both the most reactive site and the chemically preferred site for Ag/ligand (steric/electronic effects interactions between substrate and catalyst).

Subsequently, an in vitro anticancer activity test was conducted on cyclic sulfamate compounds towards MCF-7 breast cancer cells, using an MTT assay method (Table 4). All tested compounds (10 μM) exhibited some degree of inhibitory activity on breast cancer cells. It was determined that compounds **7a** had the best anticancer activity. We are in the process of investigating the anti-tumor activity of cyclic sulfamates in vivo and their mechanism of action.

## 3. Experimental Section

### 3.1. General Procedures

Unless otherwise stated, all experiments were carried out in oven-dried glassware under argon with dry solvents. All the reagents were purchased commercially and used without further purification. Dry solvents were purchased commercially. All reactions were monitored by thin-layer chromatography (TLC). TLC was performed using Huanghai 8 ± 0.2-μm precoated silica gel glass plates (0.2 ± 0.03 mm) and visualized under a UV fluorescence lamp and quenched by KMnO_4_ or phosphomolybdic acid staining. Flash chromatography was performed using Huanghai silica gel (particle size 200~300). ^1^H NMR spectra were recorded at 500 MHz, and ^13^C NMR spectra were recorded at 125 MHz using a Bruker Avance 500M spectrometer. Mass spectra were recorded on an Ultima Global spectrometer with an ESI source.

### 3.2. General Procedure for the Preparation of 4,5-Dimethyl-5-Phenyl-1,2,3-Oxathiazinane 2,2-Dioxide (***1a*** and ***1b***)

After a suspension of the **L3** (38 mg, 0.117 mmol) and AgClO_4_ (8 mg, 0.039 mmol) in dry CH_2_Cl_2_ (1 mL) was stirred in a Schlenk tube for 1 h at room temperature, protected from light with aluminum foil, a solution of the 2-methyl-2-phenylpropyl sulfamate ester **1** (0.1 g, 0.39 mmol) in dry CH_2_Cl_2_ (8.75 mL) was added. PhIO (0.3 g, 1.365 mmol) and 4 Å MS (0.37 g) were added, and the resulting solution was stirred at 55 °C for 24 h. After that, saturated aqueous NH_4_Cl (0.2 mL) was added, and the organic layer was separated and evaporated, to remove solvent under reduced pressure. The residue was subjected to column chromatography on silica gel (200–300 mesh) using PE/EA = 10/1 to 4/1 as an eluent, to produce 4,5-dimethyl-5-phenyl-1,2,3-oxathiazinane 2,2-dioxide.

### 3.3. General Procedure for the Cytotoxicity Test of Products

Cell culture: the MCF-7 cells used in this experiment were cultured in a humidified atmosphere (37 °C, 5.0% CO_2_) and grown in serum medium at a density of 6 × 10^5^ cells/dish in 25 cm^2^ cell culture flasks. MCF-7 cells were cultured in DMEM containing 10% premium fetal bovine serum (FBS) and 1% penicillin-streptomycin.

Cytotoxicity test: We investigated the cytotoxicity of the products for the MCF-7 cells using an MTT assay. The cell viability was evaluated based on the reduction of MTT to formazan crystals using mitochondrial dehydrogenases. Typically, 1 × 10^3^ MCF-7 cells in 50 μL washing buffer (Dulbecco’s phosphate buffered saline, PBS, Gibco, Shanghai, China) were pre-seeded to each test well in a 96-well plate and then incubated with DMEM for 24 h. Next, the culture medium was taken out and fresh culture medium with different products (10 μM) was added. The cells were incubated for 24 h, and then 90 μL fresh DMEM and 10 μL MTT solution were added into each well and incubated for another 0.5 h. Finally, the absorbance intensity at 490 nm was recorded using a Bio-Tek Multi-Mode Microplate Reader (Winooski, VT, USA) to assess the cell viability. All the experiments were conducted at least 3 times. In this way, cell viability measurements in MCF-7 cells were performed.

### 3.4. Density Functional Theory (DFT) Calculations

Density functional theory (DFT) calculations were performed with the Gaussian software package. We used semi-empirical methods (PM6) to calculate the probable structures for all the complexes, followed by DFT calculations to estimate their structure. Geometries were optimized using the PBEPBE functional and a mixed basis set of Lanl2DZ for Ag and 3-21G(d) for other atoms. All atoms in dichloromethane (DCM) used the SMD solvation model.

2-methyl-2-phenylbutyl sulfamate (**1**)

97% yield, yellow oil. ^1^H NMR (500 MHz, DMSO-d6) δ 7.37 (s, 2H), 7.29–7.22 (m, 4H), 7.14 (m, 1H), 4.04 (d, *J* = 9.4 Hz, 1H), 3.98 (d, *J* = 9.5 Hz, 1H), 1.67 (dt, *J* = 14.7, 7.3 Hz, 1H), 1.54 (m, 1H), 1.22 (s, 3H), 0.55 (t, *J* = 7.4 Hz, 3H). ^13^C NMR (126 MHz, DMSO-d6) δ 142.25, 126.62, 124.77, 124.53, 74.19, 39.73, 29.11, 20.04, 6.47.; HRMS (ESI-TOF^+^): *m*/*z* Calcd. for C_11_H_18_NO_3_S [(M+H)^+^]: 244.1007. Found: 244.1011.

2-(2-chlorophenyl)-2-methylbutyl sulfamate (**2**)

91% yield, yellow oil. ^1^H NMR (500 MHz, DMSO-d6) δ 7.40 (s, 2H), 7.31 (m, 2H), 7.21 (m, 2H), 4.50 (d, *J* = 9.4 Hz, 1H), 4.11 (d, *J* = 9.5 Hz, 1H), 2.13 (dd, *J* = 14.2, 7.3 Hz, 1H), 1.64 (dd, *J* = 14.1, 7.3 Hz, 1H), 1.37 (s, 3H), 0.53 (t, *J* = 7.5 Hz, 3H). ^13^C NMR (126 MHz, DMSO-d6) δ 139.27, 132.28, 131.68, 130.46, 128.42, 127.18, 73.68, 43.12, 22.65, 8.28.; HRMS (ESI-TOF^+^): *m*/*z* Calcd. for C_11_H_17_ClNO_3_S [(M+H)^+^]: 278.0618. Found: 278.0615.

2-(3-chlorophenyl)-2-methylbutyl sulfamate (**3**)

87% yield, yellow oil. ^1^H NMR (500 MHz, DMSO-d6) δ 7.38 (d, *J* = 2.3 Hz, 2H), 7.29 (dd, *J* = 5.0, 3.4 Hz, 2H), 7.24 (dt, *J* = 8.0, 1.5 Hz, 1H), 7.22–7.20 (m, 1H), 4.06 (d, *J* = 9.5 Hz, 1H), 3.97 (d, *J* = 9.9 Hz, 1H), 1.67 (m, 1H), 1.52 (dd, *J* = 14.1, 7.3 Hz, 1H), 1.22 (s, 3H), 0.55 (t, *J* = 7.4 Hz, 3H). ^13^C NMR (126MHz, DMSO-d6) δ 146.65, 133.09, 129.99, 126.44, 126.20, 125.24, 75.44, 41.58, 30.53, 7.95.; HRMS (ESI-TOF^+^): *m*/*z* Calcd. for C_11_H_17_ClNO_3_S [(M+H)^+^]: 278.0618. Found: 278.0619.

2-(4-chlorophenyl)-2-methylbutyl sulfamate (**4**)

91% yield, yellow oil. ^1^H NMR (500 MHz, DMSO-d6) δ 7.38 (s, 2H), 7.29 (d, *J* = 2.4 Hz, 4H), 4.05 (d, *J* = 9.7 Hz, 1H), 3.96 (d, *J* = 9.3 Hz, 1H), 1.65 (m, 1H), 1.52 (m, 1H), 1.21 (s, 3H), 0.54 (t, J = 7.4 Hz, 3H). ^13^C NMR (126 MHz, DMSO-d6) δ 142.90, 130.83, 128.43, 128.05, 75.55, 41.21, 30.57, 21.40, 7.93.; HRMS (ESI-TOF^+^): *m*/*z* Calcd. for C_11_H_17_ClNO_3_S [(M+H)^+^]: 278.0618. Found: 278.0618.

2-(4-(tert-butyl)phenyl)-2-methylbutyl sulfamate (**5**)

87% yield, yellow oil. ^1^H NMR (500 MHz, DMSO-d6) δ 7.36 (s, 2H), 7.28–7.24 (m, 3H), 7.19 (s, 1H), 4.01 (d, *J* = 9.4 Hz, 1H), 3.96 (d, *J* = 9.4 Hz, 1H), 1.65 (dd, *J* = 14.0, 7.3 Hz, 1H), 1.53 (dt, *J* = 13.8, 7.2 Hz, 1H), 1.20 (s, 3H), 1.19 (s, 9H), 0.56 (t, *J* = 7.4 Hz, 3H). ^13^C NMR (126 MHz, DMSO-d6) δ 148.18, 140.76, 125.97, 124.91, 75.72, 40.84, 33.97, 31.10, 30.59, 21.68, 8.12.; HRMS (ESI-TOF^+^): *m*/*z* Calcd. for C_15_H_26_NO_3_S [(M+H)^+^]: 300.1633. Found: 300.1634.

2-methyl-2-phenylpentyl sulfamate (**6**)

80% yield, yellow oil. ^1^H NMR (500 MHz, DMSO-d6) δ 7.36 (s, 2H), 7.26 (d, *J* = 6.5 Hz, 4H), 7.13 (m, 1H), 4.03 (d, *J* = 9.4 Hz, 1H), 3.97 (d, *J* = 9.3 Hz, 1H), 1.60 (m, 1H), 1.48 (td, *J* = 13.4, 12.9, 4.6 Hz, 1H), 1.24 (s, 3H), 1.05-0.96 (m, 1H), 0.86 (ddd, *J* = 19.6, 9.8, 6.0 Hz, 1H), 0.71 (t, *J* = 7.2 Hz, 3H). ^13^C NMR (126 MHz, DMSO-d6) δ 144.12, 128.18, 126.20, 126.07, 75.98, 41.13, 40.61, 22.11, 16.60, 14.46.; HRMS (ESI-TOF^+^): *m*/*z* Calcd. for C_12_H_20_NO_3_S [(M+H)^+^]: 258.1164. Found: 258.1165.

2,4-dimethyl-2-phenylpentyl sulfamate (**7**)

90% yield, yellow oil. ^1^H NMR (500 MHz, DMSO-d6) δ 7.36 (s, 2H), 7.32–7.27 (m, 2H), 7.25 (t, *J* = 7.8 Hz, 2H), 7.17–7.10 (m, 1H), 4.02–3.91 (m, 2H), 1.59 (dd, *J* = 14.0, 6.1 Hz, 1H), 1.46 (dd, *J* = 14.0, 5.4 Hz, 1H), 1.39-1.32 (m, 1H), 1.28 (s, 3H), 0.66 (d, *J* = 6.6 Hz, 3H), 0.47 (d, *J* = 6.6 Hz, 3H). ^13^C NMR (126 MHz, DMSO-d6) δ 144.17, 128.07, 126.41, 126.11, 76.46, 46.97, 41.35, 24.75, 24.28, 23.82, 22.23.; HRMS (ESI-TOF^+^): *m*/*z* Calcd. for C_13_H_22_NO_3_S [(M+H)^+^]: 272.1320. Found: 272.1322.

3-cyclopropyl-2-methyl-2-phenylpropyl sulfamate (**8**)

88% yield, yellow oil. ^1^H NMR (500 MHz, DMSO-*d*_6_) δ 7.37 (s, 2H), 7.33–7.29 (m, 2H), 7.25 (t, *J* = 7.8 Hz, 2H), 7.14 (d, *J* = 7.1 Hz, 1H), 4.15 (d, *J* = 9.4 Hz, 1H), 4.06 (d, *J* = 9.4 Hz, 1H), 1.62 (dd, *J* = 14.0, 5.9 Hz, 1H), 1.39 (dd, *J* = 14.0, 6.9 Hz, 1H), 1.32 (s, 3H), 0.34–0.15 (m, 3H), −0.04–−0.13 (m, 1H), −0.13–−0.21 (m, 1H). ^13^C NMR (126 MHz, DMSO-d6) δ 144.61, 128.11, 126.24, 126.05, 75.44, 43.51, 42.11, 22.50, 6.04, 4.79, 4.06.; HRMS (ESI-TOF^+^): *m*/*z* Calcd. for C_13_H_20_NO_3_S [(M+H^+^]: 270.1164. Found: 270.1165.

3-methoxy-2-methyl-2-phenylpropyl sulfamate (**9**)

89% yield, yellow oil. ^1^H NMR (500 MHz, DMSO-d6) δ 7.39 (s, 2H), 7.34–7.28 (m, 2H), 7.25 (dd, J = 8.5, 6.8 Hz, 2H), 7.18–7.09 (m, 1H), 4.17 (d, *J* = 9.5 Hz, 1H), 4.08 (d, *J* = 9.5 Hz, 1H), 3.40 (s, 2H), 3.15 (s, 3H), 1.23 (s, 3H). ^13^C NMR (126 MHz, DMSO-d6) δ 147.97, 133.38, 131.65, 82.50, 78.29, 63.91, 47.73, 25.94.; HRMS (ESI-TOF^+^): *m*/*z* Calcd. for C_11_H_18_NO_4_S [(M+H)^+^]: 260.0957. Found: 260.0957.

3-ethoxy-2-methyl-2-phenylpropyl sulfamate (**10**)

91% yield, yellow oil. ^1^H NMR (500 MHz, DMSO-d6) δ 7.39 (s, 2H), 7.32 (d, *J* = 7.8 Hz, 2H), 7.27–7.23 (m, 2H), 7.15 (d, *J* = 7.5 Hz, 1H), 4.19 (d, *J* = 9.4 Hz, 1H), 4.09 (d, *J* = 9.4 Hz, 1H), 3.43 (s, 2H), 3.33 (dd, *J* = 7.0, 2.1 Hz, 2H), 1.23 (s, 3H), 0.99 (t, *J* = 7.0 Hz, 3H). ^13^C NMR (126 MHz, DMSO-d6) δ 142.80, 128.11, 126.41, 126.37, 74.97, 73.12, 66.06, 42.43, 20.70, 14.87.; HRMS (ESI-TOF^+^): *m*/*z* Calcd. for C_12_H_20_NO_4_S [(M+H)^+^]: 274.1113. Found: 274.1112.

2-methyl-2-phenylhex-4-yn-1-yl sulfamate (**11**)

88% yield, yellow oil. ^1^H NMR (500 MHz, DMSO-*d*_6_) δ 7.41 (s, 2H), 7.34 (d, *J* = 7.5 Hz, 2H), 7.26 (t, *J* = 7.8 Hz, 2H), 7.15 (t, *J* = 7.2 Hz, 1H), 4.15 (d, *J* = 9.4 Hz, 1H), 4.06 (d, *J* = 9.4 Hz, 1H), 2.51-2.40 (m, 2H), 1.60 (t, *J* = 2.6 Hz, 3H), 1.31 (s, 3H). ^13^C NMR (500 MHz, DMSO-d6) δ 143.42, 128.12, 126.41, 126.26, 78.21, 75.68, 74.66, 41.09, 28.34, 22.59, 3.14.; HRMS (ESI-TOF^+^): *m*/*z* Calcd. for C_13_H_18_NO_3_S [(M+H)^+^]: 268.1007. Found: 268.1007.

2-methyl-2-phenylpent-4-yn-1-yl sulfamate (**12**)

93% yield, yellow oil. ^1^H NMR (500 MHz, DMSO-d6) δ 7.42 (s, 2H), 7.38–7.31 (m, 2H), 7.26 (t, *J* = 7.8 Hz, 2H), 7.19–7.12 (m, 1H), 4.14 (d, *J* = 9.5 Hz, 1H), 4.06 (d, *J* = 9.6 Hz, 1H), 2.72 (t, *J* = 2.6 Hz, 1H), 2.59 (dd, *J* = 16.9, 2.7 Hz, 1H), 2.48 (dd, *J* = 16.8, 2.7 Hz, 1H), 1.32 (s, 3H). ^13^C NMR (126 MHz, DMSO-d6) δ 148.28, 133.40, 131.77, 131.55, 86.22, 79.80, 78.83, 46.23, 33.10, 27.64.; HRMS (ESI-TOF^+^): *m*/*z* Calcd. for C_12_H_16_NO_3_S [(M+H)^+^]: 254.0851. Found: 254.0852.

2-methyl-2-phenylpent-4-en-1-yl sulfamate (**13**)

61% yield, yellow oil. ^1^H NMR (500 MHz, DMSO-d6) δ 7.39 (s, 2H), 7.32–7.28 (m, 2H), 7.27–7.24 (m, 2H), 7.14 (t, J = 7.1 Hz, 1H), 5.39 (ddt, J = 17.2, 10.1, 7.3 Hz, 1H), 5.01–4.84 (m, 2H), 4.05 (d, J = 9.5 Hz, 1H), 4.00 (d, J = 9.5 Hz, 1H), 2.46–2.42 (m, 1H), 2.29 (dd, J = 13.9, 7.6 Hz, 1H), 1.23 (s, 3H). ^13^C NMR (126 MHz, DMSO-d6) δ 143.74, 133.73, 128.18, 126.31, 126.23, 118.23, 75.44, 42.63, 41.01, 22.04.; HRMS (ESI-TOF+): *m*/*z* Calcd. for C_12_H_18_NO_3_S [(M+H)^+^]: 256.1007. Found: 256.1007.

4,5-dimethyl-5-phenyl-1,2,3-oxathiazinane 2,2-dioxide (**1a**)

76% yield, dr = 5.1:1, yellow oil. Major product: ^1^H NMR (500 MHz, DMSO-d6) δ 7.77 (d, *J* = 9.8 Hz, 1H), 7.43–7.30 (m, 5H), 4.63 (dd, *J* = 11.4, 0.9 Hz, 1H), 4.11 (d, *J* = 11.4 Hz, 1H), 4.10–4.02 (m, 1H), 1.43 (d, *J* = 0.8 Hz, 3H), 0.76 (d, *J* = 6.8 Hz, 3H). ^13^C NMR (101 MHz, DMSO-d6) δ 140.46, 129.25, 127.79, 127.16, 81.00, 58.11, 40.01, 14.93, 13.86.; Minor product: ^1^H NMR (400 MHz, DMSO-d6) δ 7.61 (d, *J* = 8.3 Hz, 1H), 7.52–7.45 (m, 2H), 7.29 (m, 3H), 4.75 (d, *J* = 12.0 Hz, 1H), 4.54 (d, *J* = 12.0 Hz, 1H), 4.06-4.03 (m, 1H), 3.75–3.69 (m, 1H), 1.28 (s, 2H), 0.92 (d, *J* = 6.9 Hz, 3H). ^13^C NMR (101 MHz, DMSO-d6) δ 140.93, 128.51, 128.21, 127.26, 79.81, 59.09, 38.60, 22.26, 16.29.; HRMS (ESI-TOF^+^): *m*/*z* Calcd. for C_11_H_15_NNaO_3_S [(M+Na)^+^]: 264.0670. Found: 264.0678.

5-(2-chlorophenyl)-4,5-dimethyl-1,2,3-oxathiazinane 2,2-dioxide (**2a**)

70% yield, dr > 20:1, yellow oil. ^1^H NMR (500 MHz, DMSO-d6) δ 7.56 (dd, *J* = 7.1, 2.2 Hz, 1H), 7.50 (dd, *J* = 7.3, 2.2 Hz, 1H), 7.40 (ddd, *J* = 6.9, 4.6, 2.0 Hz, 3H), 5.17 (d, *J* = 13.1 Hz, 1H), 4.80 (d, *J* = 13.1 Hz, 1H), 3.92 (t, *J* = 5.5 Hz, 1H), 1.67 (s, 3H), 1.23 (d, *J* = 10.3 Hz, 3H) ^13^C NMR (126 MHz, DMSO-d6) δ 139.78, 132.52, 131.84, 129.29, 128.09, 127.45, 74.44, 54.47, 36.30, 34.94, 22.82.; HRMS (ESI-TOF^+^): *m*/*z* Calcd. for C_11_H_14_ClNNaO_3_S [(M+Na)^+^]: 298.0281. Found: 298.0283.

5-(3-chlorophenyl)-4,5-dimethyl-1,2,3-oxathiazinane 2,2-dioxide (**3a**)

78% yield, dr = 2:1, yellow oil. Major product: ^1^H NMR (500 MHz, DMSO-d6) δ 7.84 (d, *J* = 9.8 Hz, 1H), 7.49 -7.39 (m, 5H), 4.63 (d, *J* = 11.4 Hz, 1H), 4.19 (d, *J* = 11.5 Hz, 1H), 4.13-4.05 (m, 1H), 1.45 (s, 3H), 0.81 (d, *J* = 6.8 Hz, 3H). ^13^C NMR (126 MHz, DMSO-d6) δ 142.72, 133.57, 130.56, 127.44, 126.90, 125.51, 80.09, 57.38, 39.47.14.50, 13.38.; Minor product: ^1^H NMR (500 MHz, DMSO-d6) δ 7.68 (d, *J* = 8.7 Hz, 1H), 7.58 (d, *J* = 1.9 Hz, 1H), 7.42-7.36 (m, 3H), 4.78 (d, *J* = 12.2 Hz, 1H), 4.57 (d, *J* = 12.1 Hz, 1H), 4.10–4.04 (m, 1H), 3.76 (dd, *J* = 8.4, 6.6 Hz, 1H), 1.30 (s, 2H), 0.96 (d, *J* = 6.9 Hz, 3H). ^13^C NMR (126 MHz, DMSO-d6) δ 132.88, 129.64, 127.98, 126.86, 79.31, 58.48, 38.34, 21.27, 15.65.; HRMS (ESI-TOF+): *m*/*z* Calcd. for C_11_H_14_ClNNaO_3_S [(M+Na)+]: 298.0281. Found: 298.0284.

5-(4-chlorophenyl)-4,5-dimethyl-1,2,3-oxathiazinane 2,2-dioxide (**4a**)

80% yield, dr = 3.8:1, yellow oil. Major product: ^1^H NMR (500 MHz, DMSO-d6) δ 7.80 (d, *J* = 9.8 Hz, 1H), 7.45-7.41 (m, 4H), 4.63-4.57 (m, 1H), 4.12 (d, *J* = 11.4 Hz, 1H), 4.07-4.00 (m, 1H), 1.45-1.37 (m, 3H), 0.76 (d, *J* = 6.8 Hz, 3H). ^13^C NMR (101 MHz, DMSO-d6) δ 139.52, 132.57, 129.30, 129.13, 80.64, 57.97, 39.89, 14.91, 13.84.; Minor product: ^1^H NMR (400 MHz, DMSO-d6) δ 7.56 (d, J = 8.7 Hz, 1H), 7.53–7.49 (m, 2H), 7.43–7.39 (m, 2H), 4.72 (d, *J* = 12.0 Hz, 1H), 4.53 (d, *J* = 12.1 Hz, 1H), 4.03 (dq, *J* = 9.9, 6.8 Hz, 1H), 3.72 (dd, *J* = 8.7, 6.9 Hz, 1H), 1.25 (s, 2H), 0.91 (d, *J* = 6.9 Hz, 3H). ^13^C NMR (101 MHz, DMSO-d6) δ 139.64, 132.14, 130.46, 79.90, 58.99, 16.14.; HRMS (ESI-TOF^+^): *m*/*z* Calcd. for C_11_H_14_ClNNaO_3_S [(M+Na)^+^]: 298.0281. Found: 298.0281.

5-(4-(tert-butyl)phenyl)-4,5-dimethyl-1,2,3-oxathiazinane 2,2-dioxide (**5a**)

50% yield, dr = 1.8:1, yellow oil. Major product: ^1^H NMR (500 MHz, DMSO-d6) δ 7.77 (d, *J* = 9.8 Hz, 1H), 7.45-7.34 (m, 5H), 4.63 (d, *J* = 11.5 Hz, 1H), 4.13 (d, *J* = 11.4 Hz, 1H), 4.07 (dd, *J* = 9.8, 6.7 Hz, 1H), 1.44 (s, 3H), 1.27 (s, 9H), 0.79 (d, *J* = 6.8 Hz, 3H). ^13^C NMR (126 MHz, DMSO-d6) δ 149.56, 136.92, 126.96, 126.33, 125.50, 57.57, 37.86, 31.00, 14.53, 13.43.; Minor product: ^1^H NMR (500 MHz, DMSO-d6) δ 7.59 (d, *J* = 8.4 Hz, 1H), 7.39–7.33 (m, 4H), 4.75 (d, *J* = 11.9 Hz, 1H), 4.54 (d, *J* = 11.9 Hz, 1H), 4.07 (dd, *J* = 9.8, 6.7 Hz, 1H), 3.72 (dd, *J* = 8.2, 6.7 Hz, 1H), 1.29 (s, 9H), 1.25 (s, 2H), 0.96 (d, *J* = 6.9 Hz, 3H). ^13^C NMR (126 MHz, DMSO-d6) δ 148.84, 137.39, 127.47, 124.75, 79.49, 58.66, 34.07, 21.77, 15.81, 14.53, 13.43.; HRMS (ESI-TOF^+^): *m*/*z* Calcd. for C_15_H_23_NNaO_4_S [(M+Na)^+^]: 320.1296. Found: 320.1293

4-ethyl-5-methyl-5-phenyl-1,2,3-oxathiazinane 2,2-dioxide (**6a**)

70% yield, dr > 20:1, yellow oil. ^1^H NMR (500 MHz, DMSO-d6) δ 7.68 (d, *J* = 10.1 Hz, 1H), 7.43–7.35 (m, 4H), 7.31–7.26 (m, 1H), 4.61 (dd, *J* = 11.4, 0.8 Hz, 1H), 4.08 (d, *J* = 11.4 Hz, 1H), 3.78 (m, 1H), 1.42 (s, 3H), 1.28-1.19 (m, 1H), 0.97–0.89 (m, 1H), 0.75 (t, *J* = 7.3 Hz, 3H). ^13^C NMR (500 MHz, DMSO-d6) δ 140.58, 129.28, 127.78, 127.16, 81.00, 64.78, 21.76, 14.48, 11.14.; HRMS (ESI-TOF^+^): *m*/*z* Calcd. for C_12_H_17_NNaO_3_S [(M+H)^+^]: 256.1007. Found: 256.1026.

4-isopropyl-5-methyl-5-phenyl-1,2,3-oxathiazinane 2,2-dioxide (**7a**)

75% yield, dr > 20:1, yellow oil. ^1^H NMR (500 MHz, DMSO-d6) δ 7.48–7.40 (m, 4H), 7.38–7.33 (m, 1H), 4.86 (d, *J* = 11.7 Hz, 1H), 4.38 (s, 1H), 4.08 (dd, *J* = 11.3, 4.8 Hz, 1H), 3.93 (d, *J* = 11.8 Hz, 1H), 1.63 (s, 3H), 1.29 (s, 1H), 0.97 (d, *J* = 6.7 Hz, 3H), 0.66 (d, *J* = 6.8 Hz, 3H). ^13^C NMR (126 MHz, Chloroform-d) δ 139.73, 129.04, 127.86, 126.66, 81.94, 67.16, 40.71, 28.95, 22.33, 18.65, 14.74.; HRMS (ESI-TOF^+^): *m*/*z* Calcd. for C_13_H_19_NNaO_3_S [(M+H)^+^]: 270.1164. Found: 270.1166.

4-cyclopropyl-5-methyl-5-phenyl-1,2,3-oxathiazinane 2,2-dioxide (**8a**)

80% yield, dr > 20:1, yellow oil. ^1^H NMR (500 MHz, DMSO-d6) δ 7.84 (d, *J* = 9.7 Hz, 1H), 7.47-7.28 (m, 5H), 4.72 (d, *J* = 11.5 Hz, 1H), 4.10 (d, *J* = 11.4 Hz, 1H), 3.24 (t, *J* = 9.3 Hz, 1H), 1.59 (s, 3H), 0.71 (m, 1H), 0.36-0.23 (m, 1H), 0.16 (dd, *J* = 9.8, 5.0 Hz, 1H), 0.06–−0.06 (m, 1H), −0.62 (dd, *J* = 9.7, 4.9 Hz, 1H). ^13^C NMR (126 MHz, DMSO-d6) δ 138.10, 126.24, 124.99, 124.74, 77.74, 64.84, 38.13, 12.26, 8.40, 1.66.; HRMS (ESI-TOF^+^): *m*/*z* Calcd. for C_13_H_17_NNaO_4_S [(M+H)^+^]: 290.0827. Found: 290.0828.

4-methoxy-5-methyl-5-phenyl-1,2,3-oxathiazinane 2,2-dioxide (**9a**)

81% yield, dr>20:1, yellow oil. ^1^H NMR (500 MHz, DMSO-d6) δ 8.23–8.12 (m, 1H), 7.52–7.44 (m, 2H), 7.40 (dd, *J* = 8.7, 6.8 Hz, 2H), 7.35–7.27 (m, 1H), 4.99 (dd, *J* = 8.3, 2.0 Hz, 1H), 4.50 (d, *J* = 12.0 Hz, 1H), 4.37 (d, *J* = 11.8 Hz, 1H), 3.34 (s, 3H), 1.40 (s, 3H). ^13^C NMR (126 MHz, DMSO-d6) δ 139.56, 127.99, 127.89, 126.62, 126.11, 91.00, 75.60, 55.35, 40.70, 14.90; HRMS (ESI-TOF^+^): *m*/*z* Calcd. for C_11_H_15_NNaO_4_S [(M+Na)^+^]: 280.0619. Found: 280.0613.

4-ethoxy-5-methyl-5-phenyl-1,2,3-oxathiazinane 2,2-dioxide (**10a**)

85% yield, dr > 20:1, yellow oil. ^1^H NMR (500 MHz, Chloroform-d) δ 7.32 (d, *J* = 6.1 Hz, 4H), 7.25 (m, 1H), 4.97 (d, *J* = 9.3 Hz, 1H), 4.67 (d, *J* = 12.0 Hz, 1H), 4.38 (s, 1H), 4.12 (d, *J* = 11.9 Hz, 1H), 3.75 (m, 1H), 3.37 (m, 1H), 1.45 (s, 3H), 0.93 (t, *J* = 7.1 Hz, 3H). ^13^C NMR (126 MHz, Chloroform-d) δ 137.96, 127.83, 126.78, 125.45, 90.11, 64.27, 40.57, 13.54, 13.04. HRMS (ESI-TOF^+^): *m*/*z* Calcd. for C_12_H_17_NNaO_4_S [(M+Na)^+^]: 294.0776. Found: 294.0754.

5-methyl-5-phenyl-4-(prop-1-yn-1-yl)-1,2,3-oxathiazinane 2,2-dioxide (**11a**)

78% yield, dr > 20:1, yellow oil. ^1^H NMR (500 MHz, DMSO-d6) δ 8.25 (d, *J* = 9.5 Hz, 1H), 7.44-7.34 (m, 2H), 7.31 (dd, *J* = 8.5, 6.9 Hz, 2H), 7.26-7.16 (m, 1H), 4.74 (dd, *J* = 9.6, 2.5 Hz, 1H), 4.50 (d, *J* = 11.7 Hz, 1H), 4.17 (d, *J* = 11.7 Hz, 1H), 1.59 (d, *J* = 2.4 Hz, 3H), 1.51 (s, 3H). ^13^C NMR (126 MHz, DMSO-d6) δ 139.35, 128.61, 127.48, 126.76, 82.51, 78.88, 73.62, 55.13, 15.44, 3.00.; HRMS (ESI-TOF^+^): *m*/*z* Calcd. for C_13_H_15_NNaO_3_S [(M+Na)^+^]: 288.0670. Found: 288.0671.

4-ethynyl-5-methyl-5-phenyl-1,2,3-oxathiazinane 2,2-dioxide (**12a**)

75% yield, dr > 20:1, yellow oil. ^1^H NMR (500 MHz, DMSO-d6) δ 8.50 (d, *J* = 9.7 Hz, 1H), 7.53-7.47 (m, 2H), 7.41 (dd, *J* = 8.6, 6.9 Hz, 2H), 7.34 (dd, *J* = 7.7, 1.6 Hz, 1H), 4.92 (dd, *J* = 9.6, 2.5 Hz, 1H), 4.63 (d, *J* = 11.6 Hz, 1H), 4.30 (d, *J* = 11.7 Hz, 1H), 3.38 (d, *J* = 2.4 Hz, 1H), 1.63 (s, 3H). ^13^C NMR (126 MHz, DMSO-d6) δ 138.40, 128.11, 127.06, 126.31, 78.42, 77.42, 76.89, 54.23, 14.79.; HRMS (ESI-TOF^+^): *m*/*z* Calcd. for C_12_H_13_NNaO_3_S [(M+Na)^+^]: 274.0514. Found: 274.0533.

## 4. Conclusions

In conclusion, we developed a silver/bathophenanthroline-catalyzed intramolecular amination with sulfamate esters, giving cyclic sulfamates with high site-selectivities and good yields. A variety of substrates bearing inert secondary and primary C(sp^3^)−H bonds were tolerated by this catalyst. DFT calculations further validated that the Ag/**L10** can effectively differentiate between secondary and primary C(sp^3^)−H bonds. Several in vitro experiments were conducted to evaluate the anti-tumor activity of the products. Further research of the site-selective amination of other C(sp^3^)−H bond is currently in progress.

## Data Availability

The data present in study are available in the Appendix A.

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
