# Peer review of "Silver Catalyzed Site-Selective C(sp3)−H Bond Amination of Secondary over Primary C(sp3)−H Bonds"

_molecules, 2022, doi:10.3390/molecules27196174_

Round 1

Reviewer 1 Report

The authros report on the silver(I)/phen-catalyzed C-H bond amination. Apart from some typos mentioned below, I cannot propose publication of the manuscript due to major issues.

typos:

bathophenanthroline

line 40: comparison? doesn't make sense

line 65: complexes?

major issues:

Table 2: Since CH2Cl2 boils at ~40 oC, I do not understand how the authors report on reaction at higher temperatures. Is the temperature reported here the bath temperature or the internal temperature?

the term "silver complexes" is used throughout the manuscript. However, there is no complex isolation, or even observation with spectroscopic means.

mechanism: AgL3 (i.e. 6-coordinate) complexes of Ag(I) with phenanthroline are not formed, regardless of molar ratio of reagents used. Ag(I), like other d10 metals, forms usually bis-chelates. Binuclear species may also form with bridging ligands.

Reviewer 2 Report

DFT calculation, phase of the calculation (gas or solvent) should be indicated. 

Table 6, concentration of the tested compounds should be added. 

HRMS; include H or Na to the formular that was calculated  for example compound 1 ; HRMS (ESI-TOF+): m/z Calcd. for C11H17NO3S [(M+H)+]: 244.1007. Found: 244.1011  should be  C11H18NO3S [(M+H)+]: 244.1007. Found: 244.1011

Copy of 1H-NMR, 13C-NMR spectra of the synthesized compounds are required in supporting information.

Reviewer 3 Report

This paper by Cao describes a site-selective C(sp3)−H bond amination of un-activated C(sp3)−H bonds. The authors showed that the utilization of Ph-substituted ligands is important for the high regioselectivity of the reaction. Although DFT calculations of the reactants suggest that the π-π interaction is essential, 4.9A to 6.57A may be outside the typical distances for the interactions (J. Chem. Soc., Dalton Trans. 2000, 3885). A transition-state calculation would be needed to discuss the origin of the site selectivity of the reaction. Additionally, the authors should indicate the relative stereochemistry of the major diastereomers. Therefore, I would not recommend publication of this work in Molecules at the present form. 

Additional comment: the author recheck and add recent references.

(e.g., ref 47 and ACS Catal. 2022, 12, 9, 5527–5539.)

Reviewer 4 Report

In this manuscript, the Prof. Guorui Cao and coworkers provides the Silver/ Bathophenanthrolin catalyzed the intramolecular selective amination of primary C(sp3)-H bonds and secondary C(sp3)-H bonds of sulfamate esters, this protocol give the cyclic sulfamates in good yields and high site-selectivity. The study is novel, results are impressive, and the manuscript can be considered for publication; however, there are few suggestions which should be considered for revision.

1.    Please provide the spectra of all the substrates.

2.    Most of the activated substrates are the primary and secondary C(sp3)-H bonds, what’s the result of the tertiary C-H bond reaction, such as 2,3-dimethyl-2-phenylbutyl sulfamate.

3.    Table 5, the compound of 12a:12b structure is the alkyne, check the draw style.

4.    Page 6, line 121 ‘As expected, substrates containing an alkene or propargylic substituent, show an improved preference for insertion at the allylic C-H bond activated by a neighboring π-system.’, your result did not show the alkene result, based on the 11a NMR data, it seems your substrate is alkyne, please added the result.

5.     If the 11a is alkyne, what about alkene result?

Round 2

Reviewer 1 Report

The authors revised the manuscript and it can be accepted in present form.

Reviewer 3 Report

I find the revision improved in comparison with the original submission. Overall, I think this paper could be suitable for publication in Molecules.